# Mechanical Behavior of Transparent Spinel Fabricated by Spark Plasma Sintering

Khadidja Hoggas [1,2], Salim Benaissa [3,*], Abdelbaki Cherouana [3], Sofiane Bouheroum [3], Abdenacer Assali [3], Mohamed Hamidouche [1,2] and Gilbert Fantozzi [4]

[1] Research Unit on Emerging Materials (RUEM), University of Ferhat Abbas Setif 1, Sétif 19000, Algeria; khadidja.hoggas@yahoo.com (K.H.); mhamidouche@yahoo.fr (M.H.)

[2] Institute of Optics and Precision Mechanics, University of Ferhat Abbas Setif 1, Sétif 19000, Algeria

[3] Research Unit in Optics and Photonics, CDTA, University of Setif 1, Sétif 19000, Algeria; acherouana@cdta.dz (A.C.); sbouheroum@cdta.dz (S.B.); aassali@cdta.dz (A.A.)

[4] Laboratory of Materials Engineering and Sciences (MATEIS), INSA Lyon, 69100 Villeurbanne, France; gilbert.fantozzi@insa-lyon.fr

* Correspondence: sbenaissa@cdta.dz

**Abstract:** In this work, a transparent nanostructured ceramic magnesium aluminate spinel ($MgAl_2O_4$) was fabricated by Spark Plasma Sintering (SPS) from commercial spinel nano-powders at different temperatures (1300, 1350 and 1400 °C). The sintered samples were thoroughly examined to assess their microstructural, optical, and mechanical properties. Various techniques such as SEM, AFM, spectrophotometer with an integrating sphere, instrumented Vickers indenter, Pin-on-Disk tribometer, scratch tester, and sandblasting device were employed to characterize the sintered samples. The results indicated the significant impact of the sintering temperature on the properties of the spinel samples. Particularly, the samples sintered at T = 1350 °C exhibited the highest Real In-line Transmission (RIT = 72% at 550 nm and 80% at 1000 nm). These samples demonstrated the highest hardness value (HV = 16.7 GPa) compared to those sintered at 1300 °C (HV = 15.6 GPa) and 1400 °C (HV = 15.1 GPa). The measured fracture toughness of the sintered samples increased substantially with increasing sintering temperature. Similarly, the tribological study revealed that the friction coefficient of the sintered spinel samples increased with the sintering temperature, and the spinel sintered at 1350 °C exhibited the lowest wear rate. Additionally, sandblasting and scratch tests confirmed the significant influence of the sintering temperature on the mechanical properties of the fabricated spinels. Overall, the spinel sintered at 1350 °C presented the best compromise in terms of all the evaluated properties.

**Keywords:** spinel; SPS; transparent ceramics; optical properties; mechanical properties; sandblasting





## 1. Introduction

The most commonly used transparent ceramics in industries today are single crystals and glasses. However, although highly transparent, they have rather poor mechanical properties (mechanical strength, hardness, wear resistance). In addition to being expensive, single crystals with complex shapes are hard to obtain. It is therefore important to find other appropriate alternatives. In this context, transparent polycrystalline nanostructured ceramics can be candidates, as they can offer very interesting combinations, such as: Opto-Mechanical properties, easier shaping and a possibility of large-scale production [1,2]. They are also characterized by a good thermal resistivity, a high chemical inertness and a high doping rate of active ions. These characteristics allow controlling of their optical performance. They are also optically isotropic and present a wide range of UV–Visible–IR transmission. Thanks to these favorable physical, chemical and mechanical properties, transparent ceramics can be exploited in a wide range of applications, including laser materials, scintillators, optical lenses, and even for creating transparent armor. [2]. The

manufacturing of polycrystalline ceramics with maximum optical transparency requires the elimination of all causes of light losses, such as light scattering from pores and grain boundaries. Therefore, it is necessary to use highly pure raw materials with the finest possible grain size to ensure high densities (>99.9%) and small pore sizes [3]. To meet these criteria, two main families of sintering can be implemented: (a) sintering with the application of an external macroscopic stress, including hot pressing (HP), hot isostatic pressing (HIP) and Spark Plasma Sintering (SPS), or (b) sintering without external macroscopic stress (microwave sintering).

In the last years, SPS has been shown to be a reliable alternative technique for manufacturing transparent polycrystalline ceramics at relatively low temperatures in shorter times [4]. The principle of SPS sintering is based on the application of an associated uniaxial pressure with a pulsed direct electric current. In fact, several studies confirm that the control of SPS parameters, especially heating rate, sintering temperature, holding time and pressure, allows the fabrication of fine-grained dense transparent ceramics with good optical and mechanical properties [5–9]. The application of high pressure, for example, during SPS sintering, allows good control of grain growth at lower temperatures and for shorter holding times. Sokol et al. [10] showed that the application of high pressure results in a fully dense transparent spinel, which exhibits a unique combination of high real in-line transmission (>80% at 500 nm), high Vickers hardness (up to 20 GPa) and fine grain size (~30 nm). The SPS technique has been used for many ceramic materials, namely, $ZrO_2$ [11], $Al_2O_3$ [12], $Y_3Al_5O_{12}$ [13], and MgO [14] and $MgAl_2O_4$ [5]. $MgAl_2O_4$ spinel is one of the promising transparent ceramics; it is difficult to fabricate directly from high-purity precursor powders using conventional pressureless sintering techniques [15]. However, it can be easily and qualitatively fabricated by hot pressing techniques, mainly the SPS [16]. It is used for several applications, such as transparent windows, domes, armor and optical refractories [7]. This is thanks to its important properties such as high transparency over a wide range of wavelengths (0.2–5.5 µm), relatively low density (3.58 g/cm$^{-3}$), a high level of hardness (between 16–20 GPa), high mechanical strength (150–315 MPa), high melting temperature (2135 °C) and high electrical resistivity [7].

In a potentially aggressive environment such as the Sahara, erosion of glass caused by windborne sand is a serious problem that can affect its transparency. In such conditions, spinel may be more suitable than glass for armor and protective window applications, as these types of applications require high optical and mechanical performance. The study of the resistance of a material to sand erosion can be effectively simulated in the laboratory using a sandblasting system.

Studies have shown that material removal during the erosion of brittle materials occurs primarily through the formation and propagation of radial and lateral cracks, which cause the ejection of grains from the material surface [17]. This material removal depends on several factors, such as the properties of the eroding material (shape, size, hardness, etc.) and the eroded material (hardness, toughness, etc.) as well as the conditions of exposure to erosion (speed, mass, angle of attack, etc.) [18–20]. Unfortunately, there have been only a few studies in the literature dealing with the erosion behavior of transparent ceramics. Lallemant et al. [21] and Von Helden [22] have studied the sand erosion of transparent ceramics such as alumina, $MgAl_2O_4$ spinel and different types of glass. Their results revealed that there is a strong relationship between the microstructure of the material and its resistance to erosion and alteration of its optical transmittance. Similarly, Zhou et al. [23] have found that the erosion resistance of alumina increased with the increase in hardness and fracture toughness. On the other hand, Curkovic et al. [24] showed that the erosive wear behavior of transparent alumina ceramic depends on the eroding properties (shape and hardness). Furthermore, they found that the maximum erosion occurs at an impact angle of 90°.

From all the forgoing, a transparent optical ceramic must have high mechanical properties, including sand erosion resistance and scratch resistance, in order to maintain good surface quality and high transparency. The aim of this work is the fabrication of a

dense $MgAl_2O_4$ spinel with high optical transmission and high mechanical properties for Sahara applications. For this purpose, SPS sintering of commercial spinel nano-powders at different temperatures was performed and the effects of sintering temperature on the microstructure, optical and mechanical properties of the fabricated $MgAl_2O_4$ spinel were studied. The scratch resistance and the ability to maintain good surface quality and transmission after sandblasting of the fabricated spinel are examined.

## 2. Materials and Methods

### 2.1. Sintering of Samples

A nano-powder of magnesium aluminate spinel $MgAl_2O_4$ (S25CRX 14) commercialized by Baikowski Company (La Balme-de-Sillingy, Poisy, France) was used as a raw material. Table 1 shows the impurity amounts according to the supplier's data sheet. This small impurity content has practically no effect on the properties of a transparent spinel.

**Table 1.** Impurities in the $MgAl_2O_4$ spinel powder.

| Component | Na | K | Fe | Si | Ca | S |
|---|---|---|---|---|---|---|
| Amount (ppm) | 11 | 13 | 6.5 | 14 | 6.9 | 300 |

The samples were sintered using an SPS sintering machine, type FCT System HP D25, Rauenstein, Germany. A quantity of 3.5 g of spinel powder was introduced into a graphite die with an internal diameter of 20 mm. Pellets were produced under a pressure of about 73 MPa with sintering temperatures of 1300, 1350 and 1400 °C. The heating rate was 100 °C/min up to 800 °C, then 10 °C/min up to 1100 °C and 1 °C/min up to the final sintering temperature. The change in heating rate during the SPS sintering process can have a significant influence on the density and development of the microstructure of the consolidated material. A high heating rate can lead to insufficient densification which is detrimental to the optical and mechanical properties, while too low a heating rate can extend the sintering process excessively. Thus, finding the right equilibrium of heating rate is essential to achieve the desired properties of the final material as shown by Morita et al. [25]. This sintering cycle of spinel powder was chosen based on the study of Bonnefont et al. [26] and Benaissa et al. [27].

Before characterizing the sintered pellets, both sides of the sample were polished in order to eliminate geometric imperfections and the layer contaminated by carbon of the graphite die and punch. The polishing procedure used in this study was identical to that reported in the study of [3].

### 2.2. Analysis and Characterization Techniques

The microstructures of the samples were determined using a scanning electron microscope (SEM, JSM-7001F, JEOL, Akishima, Tokyo, Japan.). The grain size was also determined from SEM micrographs using ImageJ software with a correction factor equal to 1.22 [28]. The surface topography examination was carried out using an atomic force microscope (AFM) from Asylum Research, an Oxford Instruments company, type MFP-3D, Santa Barbara, CA, USA. A Leica DCM 8 laser scanning 3D confocal microscope (the state of Hesse in central Germany) was used to measure the surface roughness (root mean square of the roughness profile (RMS)). The sample density was determined by the Archimedes method. The optical transmission was measured in the range of 200 to 1200 nm using a Shimadzu UV-1800 spectrophotometer (Kyoto, Japan). The RIT for the sample with thickness ($d_2$ = 0.88 mm), sintered at three different temperatures, was estimated using the Apetz model [28]:

$$RIT(d_2) = (1 - Rs)\left(\frac{RIT(d_1)}{1 - Rs}\right)^{\frac{d_2}{d_1}} \tag{1}$$

where $R_S$ is the total normal surface reflectance (~0.14) and RIT ($d_1$) is the real in-line transmission for the real thickness of the sample.

The total transmission (TFT), the diffuse transmission (TD), the specular reflection (*Rs*), the total reflection (RT), and the diffuse reflection (RD) were measured by a spectrophotometer type Jasco-670 (Hachioji, Tokyo, Japan), equipped with an integration sphere. The hardness measurements were performed by the instrumented Vickers indentation method, using a Zwick Roell-type apparatus, ZHU2.5 (Ulm, Germany). The Vickers hardness values obtained represent the average of three measurements taken on the samples. Vickers hardness (GPa) was calculated by the following formula [29]:

$$H_V = 0.0018544 \times (F/(d)^2) \tag{2}$$

where *F* is the applied indentation load [N] and *d* is the average length of the diagonals of the indentation imprint [mm].

Young's modulus was determined using ultrasonic testing in a single experiment [30]. The fracture toughness ($K_{IC}$) of the transparent spinel was measured using the indentation technique on specimens with polished surfaces, with six measurements taken on samples under a 5 N load. The measured radial crack lengths (2C in m) were used to determine the fracture toughness by the equation of Anstis et al. [31], which recently concluded that that the following formula provides the most accurate characterization of ceramics. [32]:

$$K_{IC} = 0.016(\frac{E}{H})^{1/2} \times \left( \frac{F}{C^{3/2}} \right) \tag{3}$$

where *H* is the Vickers hardness (GPa) and *E* (GPa) is the elastic modulus.

The tribological tests were carried out using a CSM-type Pin-On-Disk Tribometer (Version 4.5.Q, Peseux, Switzerland) with an alumina ball (Ø 6 mm) as a pin rubbing against sintered samples as disks.

Each sample was rubbed under a normal force of 10 N with a sliding radius of 6.2 mm, a linear speed of 100 rpm and a test duration of 15 min. All the tribological tests were performed in a dry environment at a room temperature of 20 °C and humidity of ~40%. During all the tests, the friction coefficient was measured as a function of time. In order to evaluate the wear rate, the width of the wear track was measured using an optical transmission microscope.

For the sandblasting tests, the transparent spinels were eroded using a horizontal type of sandblasting device [33]. The latter complies with DIN 50 332 standards and ASTM G76. A natural sand from the region of El-Oued (Southeast of Algeria) was used as eroding material. Figure 1 showed that this sand has various grains with different shapes and with an elongated index (Ei = Dmax/dmax) of 1.44. It has a fairly homogeneous particle size ranging between 250 and 500 μm with an average particle size of 374 μm (Figure 2). The chemical composition of this sand was indicated in Table 2. The fabricated spinel samples were sandblasted on one side under the following conditions:

- Projected mass: m = 200 g
- Air flow speed: v = 30 m/s
- Angle of impact: α = 90°
- Distance between the nozzle outlet and the sample: x = 50 mm.

**Table 2.** Chemical composition of the eroding sand used.

| Oxides | $SiO_2$ | $Al_2O_3$ | CaO | $K_2O$ | $SO_3$ | $Fe_2O_3$ | MgO | SrO |
|---|---|---|---|---|---|---|---|---|
| Amount (wt.%) | 90.8 | 3.86 | 1.36 | 1.32 | 1.22 | 0.592 | 0.459 | 0.0140 |

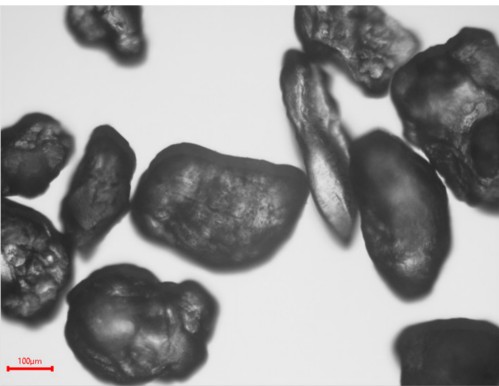

**Figure 1.** Micrograph of the sand particles used as projectiles impacts.

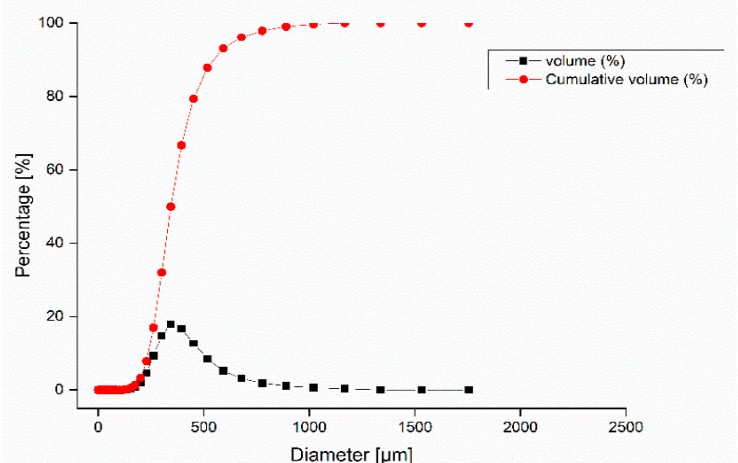

| Medium diameter | 374.66 µm |
|---|---|
| D (0.1) | 236.72 µm |
| D (0.5) | 344.63 µm |
| D (0.9) | 548.55 µm |

**Figure 2.** Particle-size distribution of the eroding sand used.

The difference in surface roughness of the sample before and after sandblasting may reflect its rate of damage following its exposure to erosion. For this purpose, roughness parameters were measured before and after sandblasting, using a Leica DCM 8 3D laser scanning confocal microscope. Two main parameters were chosen to evaluate the roughness based on this test: the root mean square roughness (Rq) according to ISO 4287 and the root mean square height (Sq) according to ISO 25178.

The progressive linear scratch test was performed using a CSM Revetest scratch device equipped with a Rockwell diamond indenter with a radius of 200 µm under a progressively increasing load from 1 to 60 N. The loading rate was 59 N/min. The scratch length was 5 mm. During all the tests, the penetration depth, the applied load and the acoustic emission were recorded to assess the characteristics of cracks and determine the critical forces. Finally, the scratches were examined by light microscopy.

## 3. Results and Discussion

### 3.1. Microstructural Characterization

SEM micrographs of the spinel samples sintered at 1300, 1350 and 1400 °C are presented in Figure 3a–c. It is clear that all the samples have a dense and homogeneous microstructure. However, it is found that the sintering temperature has a significant effect on the microstructure of the fabricated spinel, where the grain size of the samples increased significantly with increasing sintering temperature. The sample sintered at 1300 °C has the smallest average grain size of around 685 nm; this size increased to 1.9 µm in the sample sintered at 1350 °C, and reaches its maximum in the sample sintered at 1400 °C, which has the largest average grain size of about 5.3 µm. The defects that appeared on the images of

1350 and 1400 °C perhaps were caused only by the chemical treatment that we carried out before the observation at the SEM, as shown by Benaissa et al. [26].

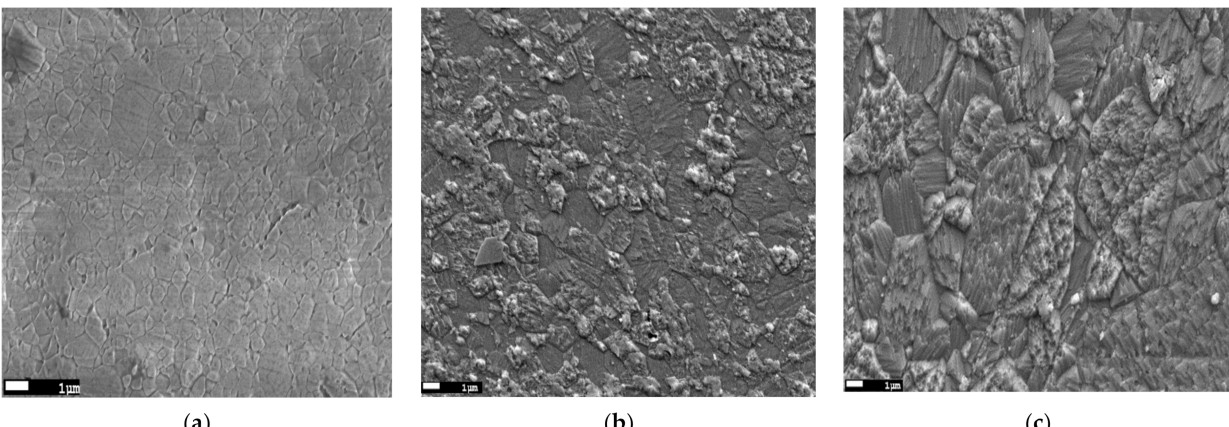

**Figure 3.** SEM micrographs of the samples sintered by SPS at: (**a**) 1300 °C; (**b**) 1350 °C and (**c**) 1400 °C.

The 2D and 3D surface morphologies of the samples sintered at different temperatures are presented in Figure 4. The effect of the sintering temperature is clearly visible on the morphology of the fabricated samples, where the grain size increased with the increase in the sintering temperature, which confirms the SEM observations (Figure 3). Furthermore, the average surface roughness RMS values obtained by AFM, in the same scanning area (25 μm$^2$), were $5.8 \pm 0.74$, $8.2 \pm 2.03$ and $61.9 \pm 5.06$ nm for the samples sintered at 1300, 1350 and 1400 °C, respectively. The increase in roughness can be mainly explained by the increase in grain size by coalescence of neighboring grains due to the increase in sintering temperature.

The relative density of the sintered samples was determined according to the Archimedes principle. The density values were $99.65 \pm 0.1\%$, $99.74 \pm 0.11\%$ and $99.86 \pm 0.03\%$ for the samples sintered at 1300, 1350 and 1400 °C, respectively. The obtained values are the means of five measurements per sample. The slight increase in density with increasing sintering temperature could be explained by the coarsening of grains and promoting of densification by sliding and rearrangement of grains, because the grains bind or weld together as they grow, which can lead to consolidation of the material. Consolidation can cause the relative density to increase because the grains are more tightly bound and the space between them is reduced [34].

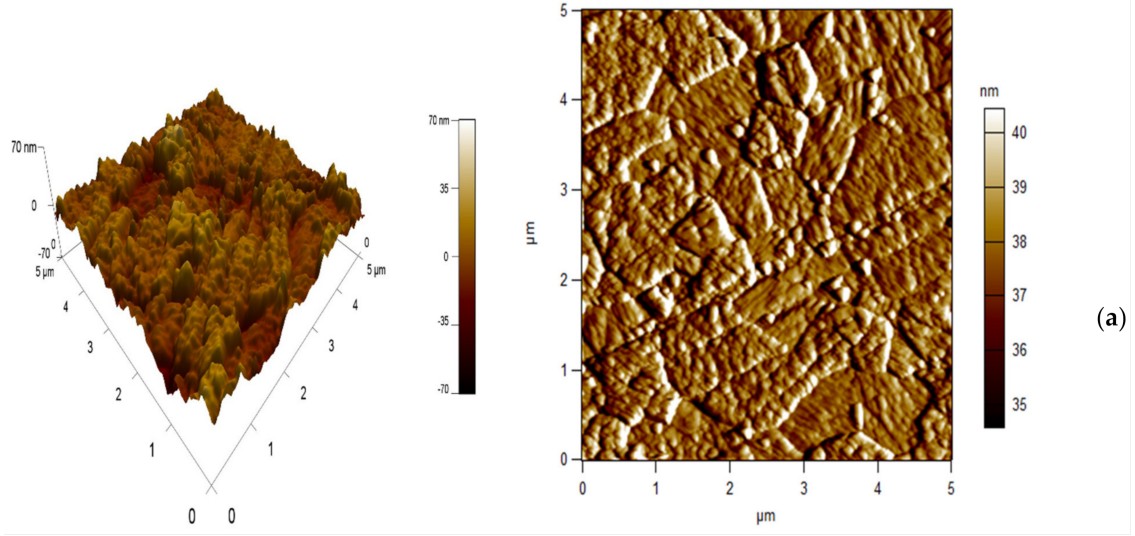

**Figure 4.** *Cont.*

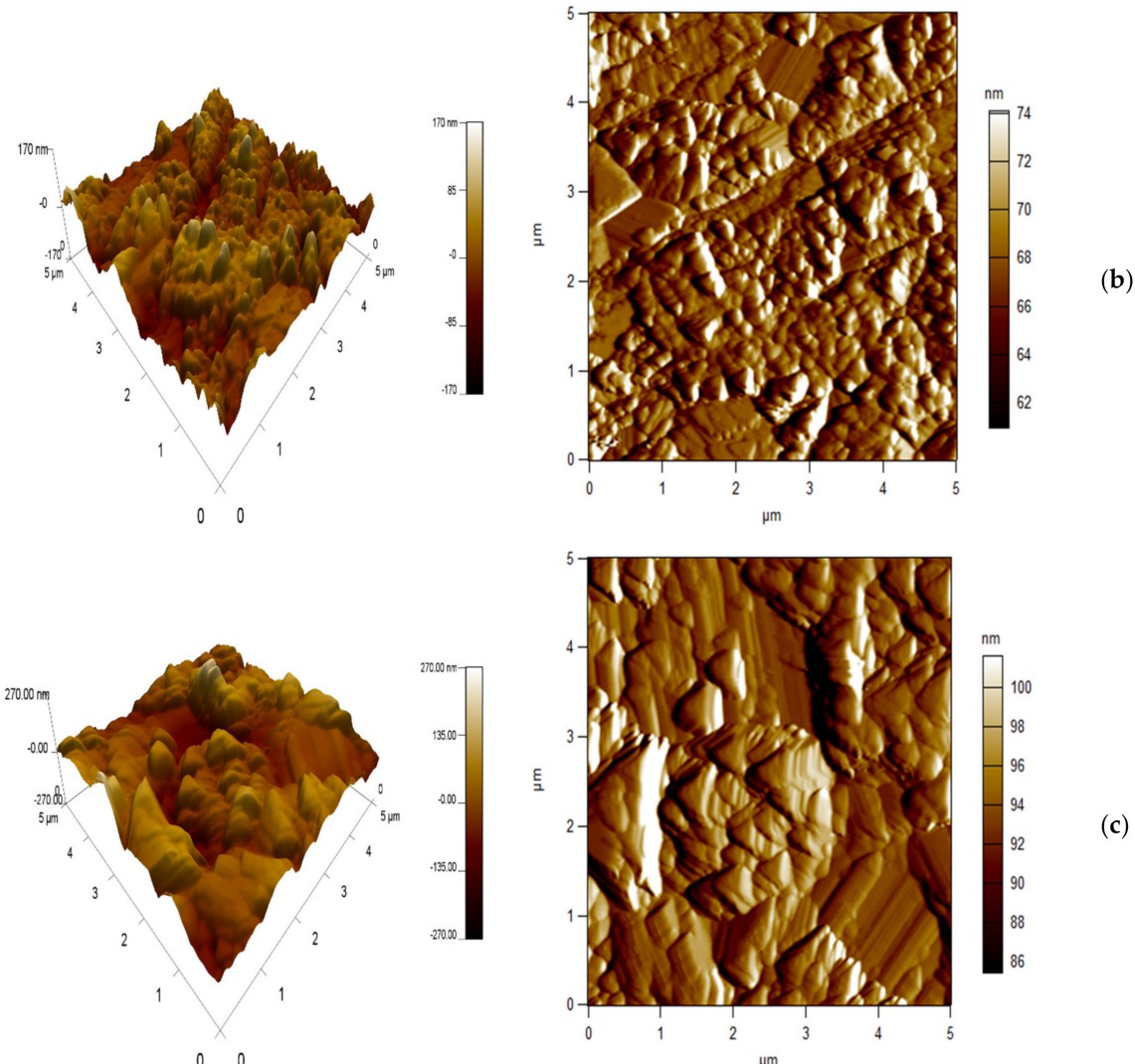

**Figure 4.** 2D and 3D AFM images of the samples sintered at: (**a**) 1300 °C, (**b**) 1350 °C and (**c**) 1400 °C.

### 3.2. Optical Characterization

Figure 5 shows the spectra of real in-line transmission of the samples sintered at different temperatures. It can be observed that the samples sintered at 1300 and 1350 °C have almost the same high RIT in the visible range (400–800 nm) and that the best optical real in-line transmission is reached with samples sintered at 1300 and 1350 °C. This RIT decreased abruptly in the sample sintered at 1400 °C, which has a very low transmission. The decrease in the transmission at high temperature can be explained by the strong contamination by carbon during sintering; Hammoud et al. [35] showed that a contamination occurred by carbon clusters originating from the powder and the environment of the SPS (graphite foil, die and felt). A high SPS temperature may further favor the diffusion from the carbon clusters, which explains the increased light absorption observed in samples sintered at 1400 °C. However, the carbon contamination can be reduced even slightly by a thermal annealing under air, as shown by Zegadi et al. [3]. Not only carbon contamination but also grain size may be an important parameter influencing transmission, as shown by Rothman et al. [36]. It should be noted that our results regarding transmission as a function of SPS temperature are in good agreement with those in the literature [37,38].

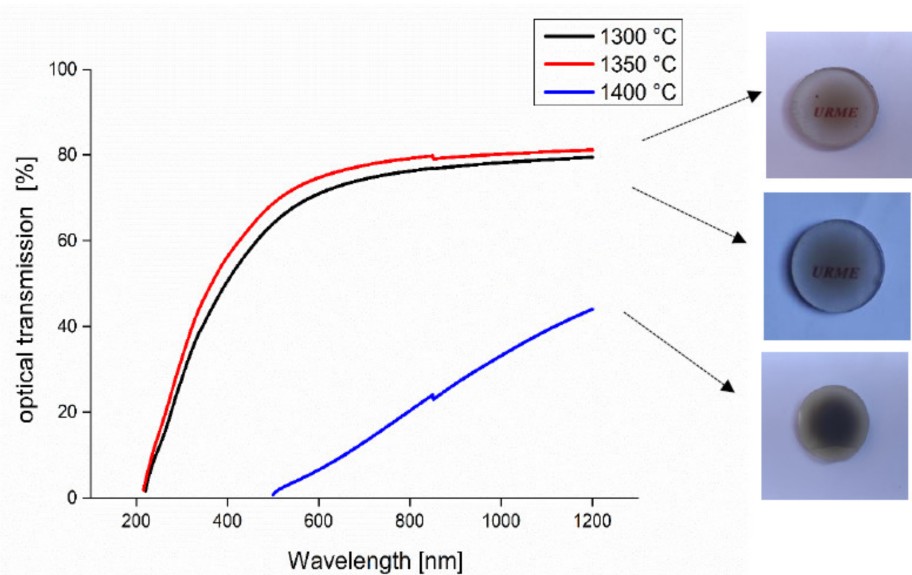

**Figure 5.** The Real In-line Transmission spectra of samples sintered by SPS at 1300, 1350 and 1400 °C.

Table 3 summarizes the RIT values recorded at 550 and 1000 nm wavelengths after correction of sample thicknesses to 0.88 mm. It can be noticed that the samples sintered at 1350 °C have the highest value of transmission (RIT = 72% at 550 nm and RIT = 80% at 1000 nm). These results are almost higher than those obtained by Benaissa et al. [27], who found an RIT = 45% at a wavelength of 550 nm and RIT = 67% at 1000 nm by SPS sintering of S30CR at the same temperature of sintering.

**Table 3.** RIT of the $MgAl_2O_4$ spinel sintered by SPS at different temperatures ($d_2$ = 0.88 mm).

| SPS Temperature | 1300 °C | 1350 °C | 1400 °C |
|---|---|---|---|
| RIT (%) at 550 nm | 68.1 | 72.3 | 3.4 |
| RIT (%) at 1000 nm | 78.2 | 80.2 | 33.2 |

To better understand the effect of sintering temperature on the different optical properties of the fabricated transparent spinel, we present the intensity of transmitted, reflected, absorbed and scattered light in Figure 6 as a function of three different wavelengths. The first conclusion that can be drawn is that the RIT increased with the increase in the wavelength, while the absorbed intensity, the diffuse transmission and the diffuse reflection decreased. The second conclusion is that the sample sintered at 1350 °C exhibited the best transmission at any wavelength, whereas the sample sintered at 1400 °C exhibited the lowest transmission at any wavelength. The poor optical transmission of the sample sintered at 1400 °C can be explained by carbon contamination during sintering, where this sample has been identified as a potential cause of a decrease in optical properties [35,39,40].

*3.3. Mechanical Characterization*

The variation of Vickers hardness of the sintered specimens as a function of indentation load is illustrated in Figure 7. It can be clearly seen that the hardness of the samples sintered at 1350 °C is higher than of those sintered at 1300 and 1400 °C, regardless of the indentation load. The observed variation of the hardness of the samples sintered at different temperatures is attributed mainly to the variation of their porosity and grain size (Hall and Petch law) [10]. This is because the reduction in grain size limits the propagation of dislocations, which strengthens the material and leads to an increase in its hardness. Conversely, samples with larger grain sizes demonstrate the lowest hardness values. The larger grain size promotes dislocation propagation, weakening the material and leading to a decrease in hardness.

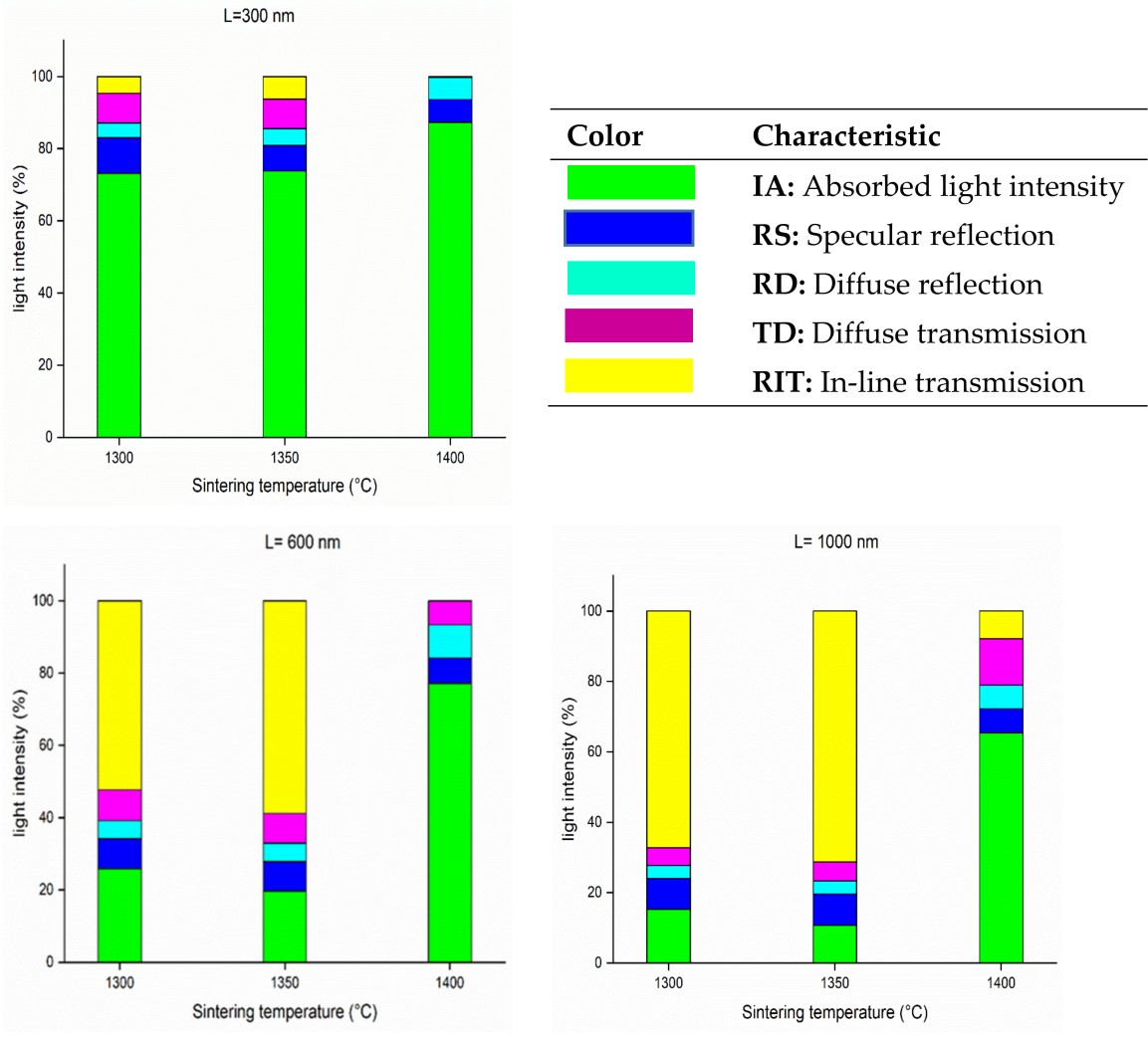

**Figure 6.** Distribution of incident light intensity within the fabricated spinel pellets.

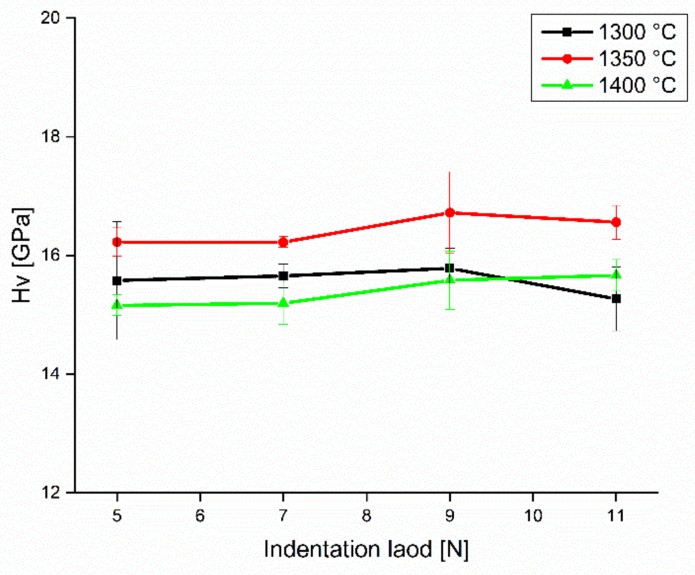

**Figure 7.** Vickers hardness as function of indentation load of the spinel sintered at 1300, 1350 and 1400 °C.

It should be noted that, in the case of spinel, the homogeneity of the microstructure, in particular the grain size, could have a more important effect on the mechanical properties than the porosity rate and the presence of inclusions [41]. From Figure 7, it can also be seen that the hardness values do not depend on the indentation load; this is due to the uniformity of the microstructure and the high relative density of the prepared spinel pellets. On the other hand, the obtained Vickers hardness values were always higher than 15.5 GPa, regardless of the sintering temperature. It should be noted that this hardness value with the same measurement method is slightly higher than that found by Krell et al. [42] (HV = 15 GPa), and considerably higher than that found by Benaissa et al. ($H_V$ = 15 GPa at 1350 °C) [27].

Table 4 show the results of elastic values obtained by the ultrasonic method for the spinel samples sintered at 1300, 1350 and 1400 °C. A gradual increase in Young's modulus with the increase in sintering temperature can be clearly observed, where the elastic modulus goes from 256 to 285 GPa by increasing the sintering temperature from 1300 to 1400 °C. This increase can be explained mainly by the microstructural changes and the increase in density with the increase in the sintering temperature, where it is well known that the elastic modulus is sensitive to the porosity.

**Table 4.** Elastic modulus and fracture toughness of the spinel samples.

| Sintering Temperature | 1300 °C | 1350 °C | 1400 °C |
|---|---|---|---|
| Elastic modulus [GPa] | 265 | 272 | 285 |
| Fracture Toughness [MPa$\sqrt{}$m] | 1.5 ± 0.1 | 1.7 ± 0.3 | 2.5 ± 0.4 |

The results of this study indicate that in order to improve the mechanical properties of spinel, a homogeneous and fine microstructure must be maintained. However, in the case of low temperature sintering (1300 °C), spinel has a slight porosity and a fine microstructure. In contrast, spinel sintered at intermediate temperature (1350 °C) is dense with a homogeneous microstructure and slightly increased grain size. Finally, in the case of sintering at a high temperature (1400 °C), the spinel presents a coarse microstructure.

The fracture toughness ($K_{IC}$) calculated from Equation (3) for an indentation load of 5 N was presented as a function of sintering temperature in Table 4. It can be noticed that the values of the fabricated spinel pellets increased gradually from 1.5 MPa$\sqrt{}$m for 1300 °C to 2.5 MPa$\sqrt{}$m when the samples were sintered at the temperature of 1400 °C.

This increase in fracture toughness is mainly due to the increase in grain size with increasing sintering temperature. In fact, the improved grain bridging increases with the grain size; this grain bridging is behind the improvement of the fracture toughness of the fabricated ceramic. It should be noted that our best fracture toughness value of the sample sintered at T = 1400 °C ($K_{IC}$ = 2.5 MPa$\sqrt{}$m) was higher than that reported by Tokariev et al. ($K_{IC}$ = 1.8 MPa$\sqrt{}$m) [43] and Benaissa et al. ($K_{IC}$ = 2.2 MPa$\sqrt{}$m) [27]. Moreover, it was slightly lower than that reported by Von Helden ($K_{IC}$ = 2.8 MPa$\sqrt{}$m) [22] under almost the same test conditions. This proves that our prepared spinel specimens exhibit a very accepted fracture toughness.

Figure 8 shows the evolution of the friction coefficient as a function of time for the spinel samples sintered by SPS at different temperatures. It can be observed that the sintering temperature has a significant influence on the tribological behavior of the fabricated spinel. The sample sintered at 1350 °C presented the most stable and the lowest coefficient of friction with a value of about 0.35. The friction coefficient increased to about 0.42 and became less stable in the sample sintered at 1300 °C.

The sample sintered at T = 1400 °C presented the most fluctuating and the highest friction coefficient with an average value of about 0.47. Thus, it can be concluded that the spinel sample sintered at 1350 °C exhibits the best tribological behavior of the friction reduction among the fabricated samples. The friction coefficient is an extrinsic material property that can be influenced by the conditions surrounding the material (temperature, humidity, lubrication, etc.) as well as the intrinsic material properties such as hardness,

roughness and especially the phase composition of the material. However, hardness and roughness seem to be the most influential factors on the friction coefficient in this study, where the sample sintered at 1350 °C, which has the highest hardness, and relatively low roughness, has the lowest friction coefficient value and showed the best frictional behavior.

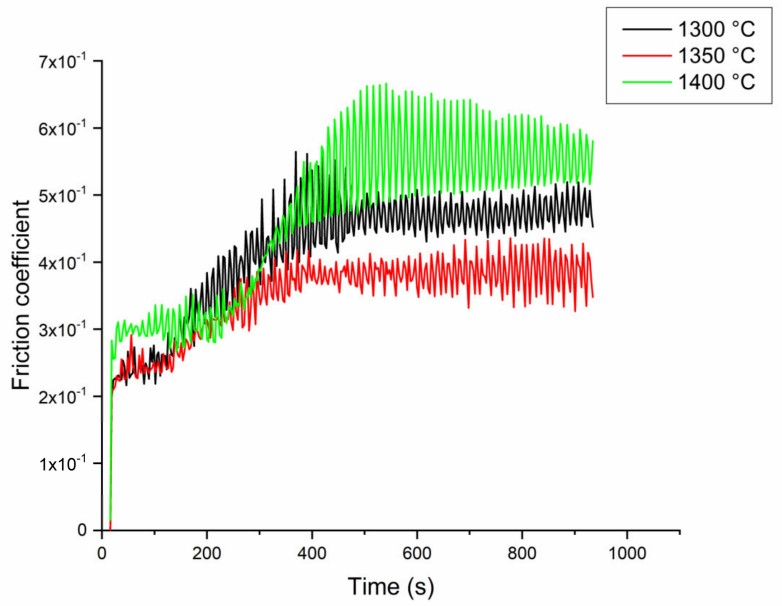

**Figure 8.** Evolution of friction coefficient as a function of time for the spinel sintered at different temperatures.

These results show that the friction coefficient depends on the one hand on hardness, and on the other hand on roughness linked to the grain size. The sample sintered at 1400 °C exhibits a large grain size (5.3 μm) and therefore a high roughness (62 nm) and the lowest hardness, giving a high friction coefficient. These results confirm those found by other researchers [27].

The width of the wear tracks can be an indication about the wear rate and the wear resistance of the materials subjected to rubbing. Figure 9 illustrates the variation in the width of the wear tracks of the samples sintered by SPS at different temperatures that underwent the wear test. The values presented are the average results of seven measurements, taken with the utmost care to ensure consistency in the experimental conditions and to ensure the reliability of the data. It is evident that the sample sintered at 1350 °C exhibits the narrowest wear track width (116 μm), albeit with a relatively high standard deviation, which could be attributed to potential surface flatness defects or inadequate cleaning of the ball during the test. The sample sintered at 1400 °C, in contrast, exhibited the widest wear (133 μm) track and the lowest wear resistance. This tribological behavior of our manufactured spinels can be explained firstly by the variation in the friction coefficient, where the spinel sintered at 1350 °C has the lowest friction coefficient and showed the highest wear resistance, while that sintered at 1400 °C has the highest friction coefficient and showed the lowest wear resistance. The tribological behavior of our materials can also be explained by the hardness variation, where the material with the highest hardness value presented the lowest wear rate, and vice versa, this corresponds to the Archard wear equation [44], which demonstrates an inverse relationship between the hardness of a material and its rate of wear.

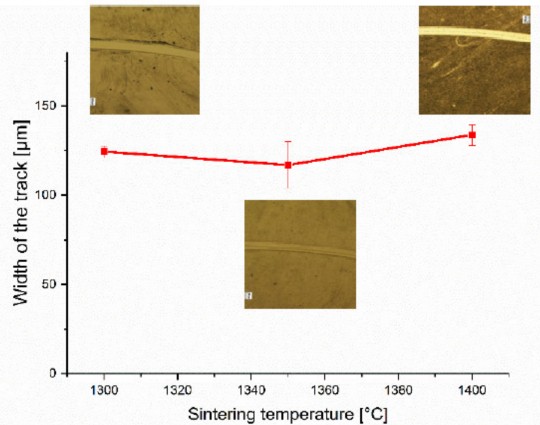

**Figure 9.** Variation in the track width as a function of sintering temperature.

### 3.4. Sandblasting

The erosion behavior of brittle materials such as polycrystalline ceramics is controlled by the formation of lateral cracks parallel to the surface. These cracks easily propagate to form a network of cracks propagating through grain boundaries. Repeated shocks by sand particles will easily remove surface material with grain ejection from the top layer. The volume of the removed material depends on the material fracture toughness and hardness [23]. As these latter values are different for each sample (Figure 7 and Table 4), one would expect different erosion rates. For the microscopic observations of defects, an optical microscope is used. It allows the surface defects to be observed and photographed and then the damage rate of a transparent ceramic surface to be seen (Figure 10). From these micrographs, it can be noticed that the sample's surface is completely eroded, and the sandblasting defects resulting from micro-fragmentation, particularly lateral cracks, exhibit a random distribution across the entire exposed surface. In addition, it is noticed that the damage worsens as the sintering temperature decreases. Thus, it is found that the spinel sintered at 1300 °C is more damaged compared to the sample sintered at 1400 °C, due to its low fracture toughness. The damage rate corresponding to the samples sintered at 1300, 1350 and 1400 °C is equal to 46.7%, 34.6% and 32%, respectively. In order to examine the mechanism of erosion and surface damage, the top view of uneroded and eroded surfaces of sintered samples was observed by confocal microscope (Figure 11). This finding confirms that, after sandblasting, the ejection of the grains from the surface layer leaves hollows in the samples surface and increases the roughness, which subsequently influences the surface reflectance Rs, as already mentioned in literature [24]. It is also noticed that the samples sintered at 1400 °C have deeper defects than the other samples because of the coarse microstructure (grain size of around 5.3 μm), and the surface state becomes more and more damaged and rough.

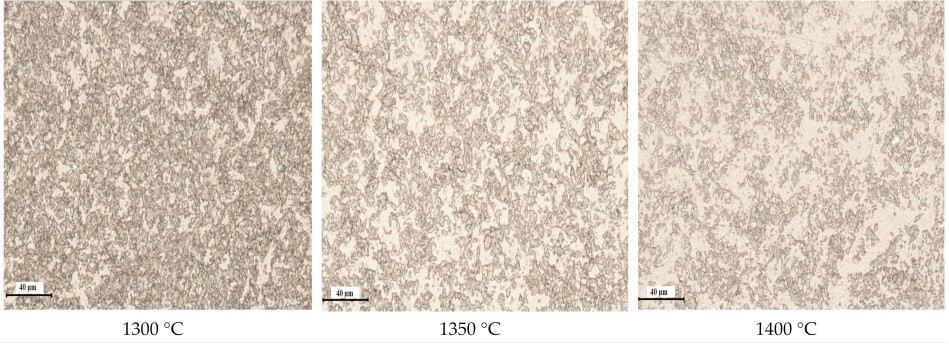

**Figure 10.** Microscopic observations of the surface defects for three sintering temperatures after sandblasting.

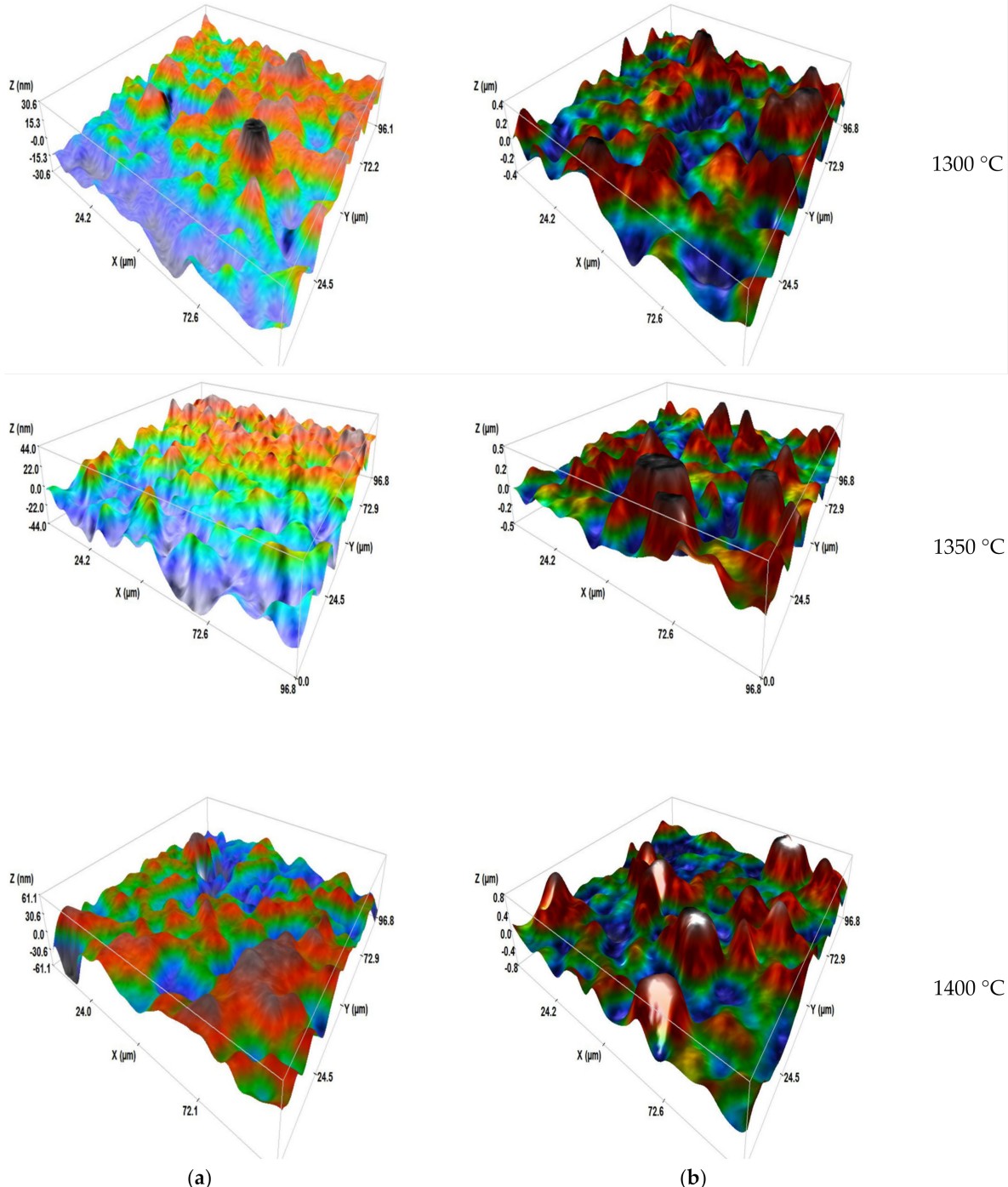

**Figure 11.** 3D images of confocal microscope for sintered sample surfaces, (**a**) before and (**b**) after the erosion.

The changes in the roughness value quadratic height (Sq: extended surface roughness parameter relating to the area) of different transparent samples before and after erosion are shown in Figure 12. It is observed that the roughness in the raw state is low in the case of samples sintered at 1300 and 1350 °C (75 and 73 nm). These results show that the samples undergo good polishing with fine microstructures; except in the case of spinel sintered at 1400 °C, which has a high Sq of the order of 125 nm. Furthermore, when the roughness results after sandblasting are examined, it can be noticed that the values of Sq increase from 75 nm up to 134 nm for the sample sintered at 1300 °C. For spinel sintered at 1350 °C, the Sq reaches 139 nm. As for spinel sintered at 1400 °C, the roughness reaches 167 nm.

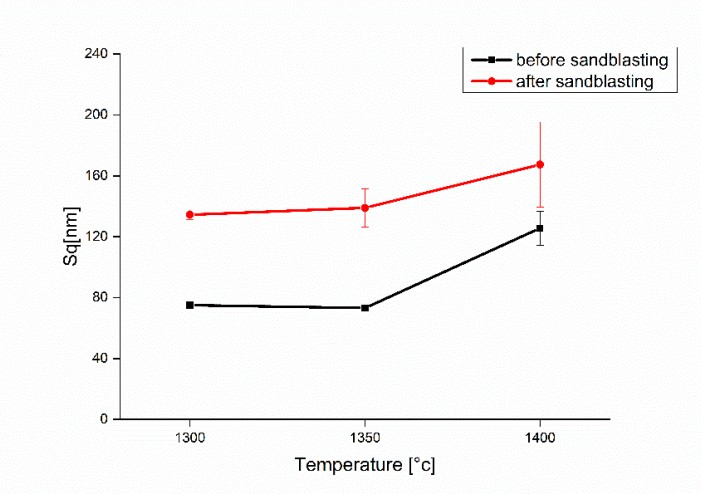

**Figure 12.** Variation of the roughness of the sintered samples before and after sandblasting.

The surface state is influenced by the defects generated by the grains of sand and the microstructure. In this case, it is found that the spinel with a larger microstructure (sintered at 1400 °C) exhibits higher roughness values than the sintered spinel with a fine microstructure (sintered at 1300 °C). The microstructure effect on the morphology of eroded surface can be explained by treating the sintered sample's surface with sand particles. The removed grains leave marks corresponding to their sizes. As a result, the roughness of the sanded surface increases with the increase in size of the removals. In a study subsequently carried out by Mroz and his collaborators [45] on the erosion of the nanostructured spinel ($MgAl_2O_4$) by sand and rain, the nano-spinel with a grain size of around 345 nm exhibits better resistance to erosion when compared with the fine grain-size spinel of 970 nm.

Figure 13 shows the optical transmission spectra of transparent spinel eroded by a mass of sand of 200 g. The spinel spectra have the same appearance, but the level of the curves differs. This shift is due to the degree of the surface damage, which scatters the incident light. It is clearly seen that the initial state (before sandblasting) is placed at the highest level regardless of the sintering temperature. After sandblasting the samples show a drop in optical transmission of about 7% at $\lambda = 550$ nm for the sintering at 1300 °C and 1350 °C. This loss of optical quality is due to the increased reflection of light from the surface, which is caused by defects induced by sand impacts.

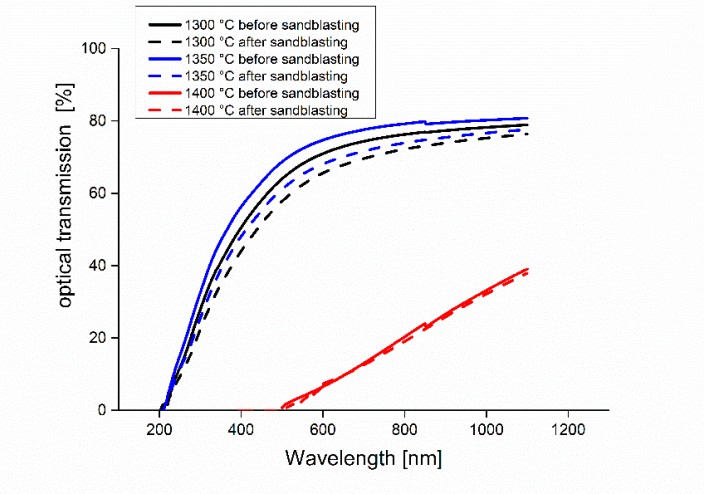

**Figure 13.** Variation of optical transmission of sintered spinel at different temperatures before and after sandblasting.

During the progressive scratching tests, it was noticed that when there is an increase in the applied normal load, three successive damage regimes occur [46] (Figure 14): first, a micro-ductile regime where there is a permanent furrow without visible damage (a); second, a cracking regime which is characterized by the initiation of a first radial crack under the effect of a normal critical load (b); and third, a micro-abrasive regime, caused by the normal breaking load (c). In this latter case, the damage becomes more important, and the formation of strong lateral cracks induces spalling, which can cause a removal of material.

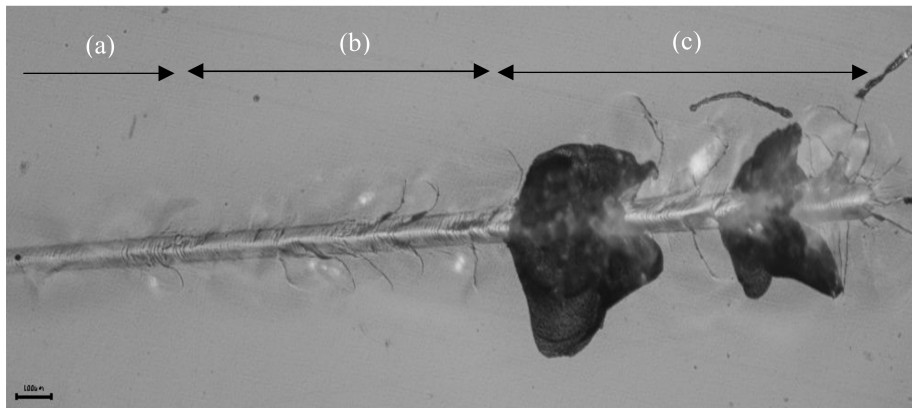

**Figure 14.** Microscopic observations of the scratching of spinel sintered at 1300 °C: (**a**) Micro-ductile regime; (**b**) Cracking regime; (**c**) Micro-abrasion regime.

From the optical observations, the normal critical cracking loads and the normal critical damage loads for the samples sintered at different temperatures are determined (Table 5).

**Table 5.** Critical forces of the scratch test for the different sintered samples.

| Critical Force | FC1 (N) | FC2 (N) |
|:---:|:---:|:---:|
| 1300 °C | 35.2 | 49.8 |
| 1350 °C | 30.9 | 42.5 |
| 1400 °C | 19.5 | 34.5 |

Sample S25CRX 14 sintered at 1400 °C has lower critical load values than the other two samples sintered at 1300 and 1350 °C. This behaviour is explained by the lower hardness of the sample sintered at a higher temperature due to the less fine-grained microstructure. Figure 15 illustrates the scratching behaviour of the different samples according to the sintering temperature. The increase in scratching force leads to large spalling areas when the sintering temperature is higher (1400 °C). The primary damage mechanism during spinel scratching is a brittle fracture. This type of behaviour has also been observed in the case of alumina [47–49].

During the scratch test, the system records the depth of penetration (Pd) and the acoustic signals (AE) associated with cracking or chipping. Figure 16a–c presents the variation of penetration depth and the acoustic emission recorded during scratching of transparent spinels, sintered at different temperatures. When the microstructure is finer, and sintering is at 1300 °C, the beginning of the acoustic emission takes place at a higher normal load. As the applied load increases and spalls form, the emission signals become more intense. Indeed, as the normal load increases, the residual penetration depth increases. At the end of the scratch test, we noted the depth reached as follows: 20.6, 7.5 and 14.2 μm, respectively, for the samples sintered at 1300, 1350 and 1400 °C.

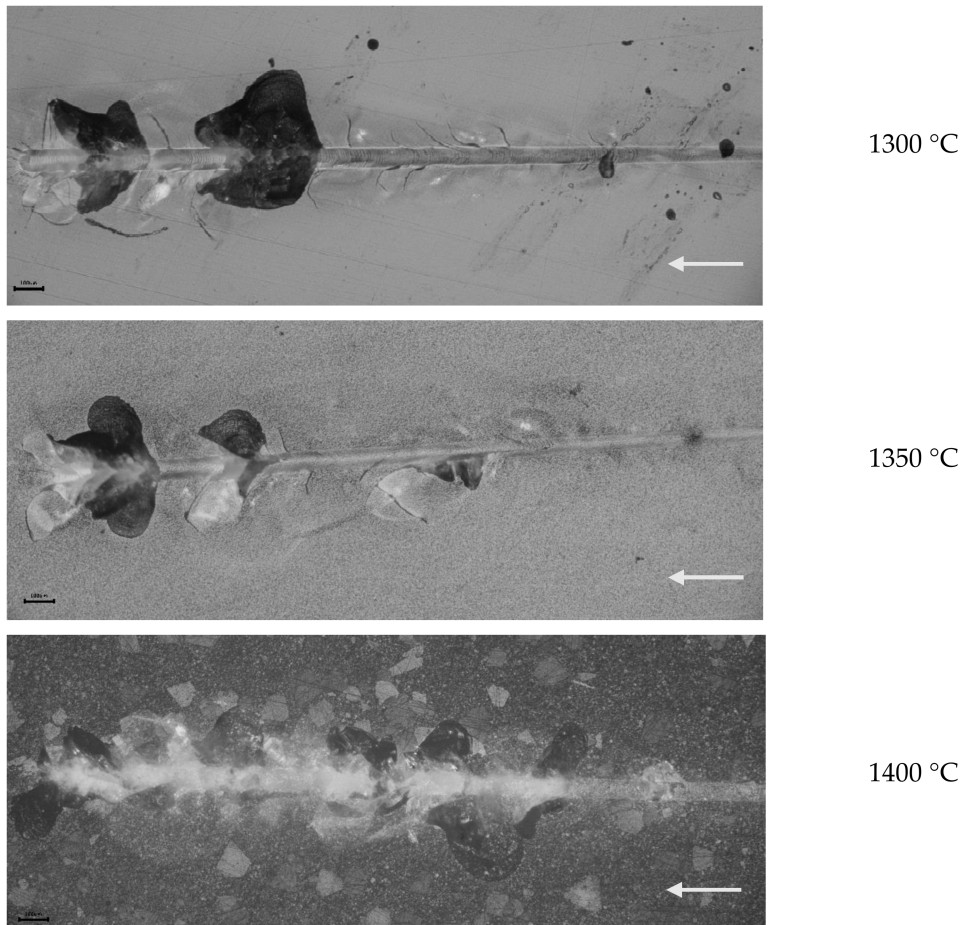

**Figure 15.** Microscopic observations of the parts of a scratch made on different transparent spinel (arrow: scratching direction).

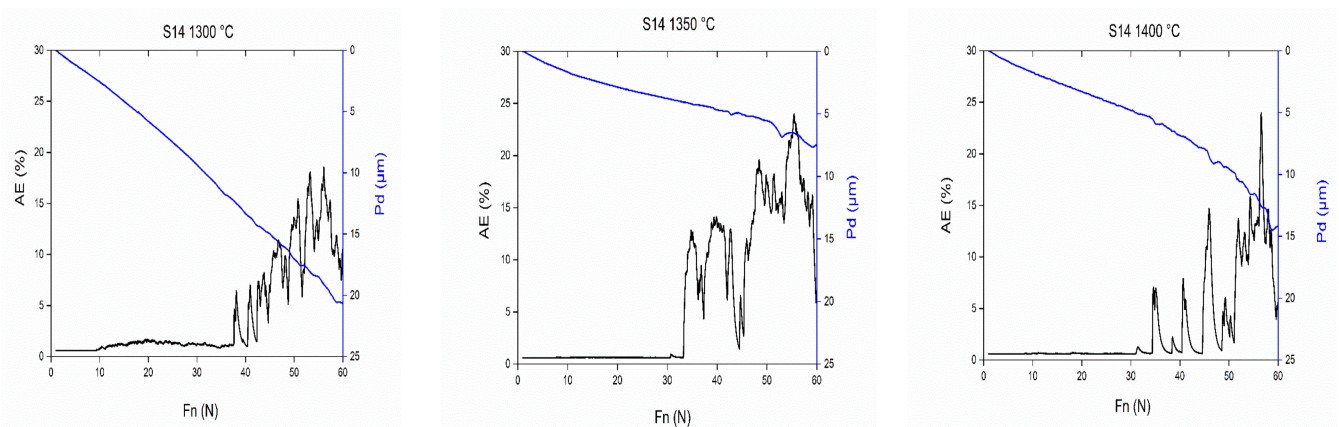

**Figure 16.** Recorded acoustic emission activities and corresponding penetration depth as a function of applied normal load for samples sintered at different temperatures.

## 4. Conclusions

The present study demonstrates the significant impact of SPS sintering temperature on the mechanical and optical properties of transparent spinel. High sintering temperatures lead to an increase in carbon contamination and promote the coalescence of residual pores, leading to the degradation in the optical properties. Conversely, it was observed that the best mechanical properties were achieved at low sintering temperatures, attributable to the presence of a small average grain size.

The main conclusions from this study are summarized as the following:

– The use of commercial nano-powder of spinel S25CRX 14, having a high chemical and crystallographic purity, results in dense (>99.9) and nanostructured-sintered pellets. It was also revealed that the optimal optical properties values found for samples sintered at 1350 °C, at a wavelength of 550 nm, are: (1) a real in-line transmission of 72.3%, (2) a total reflection RT of 13.1%, and (3) an absorbed light intensity of 23.5%.

– The samples sintered at 1350 °C led to the best compromise for mechanical properties, in particular, Vickers hardness (Hv ≈ 16 GPa), fracture toughness ($K_{IC}$ ≈ 1.4 MPa $\sqrt{m}$), and elastic modulus (E ≈ 272 GPa).

– The tribological study of the transparent spinel found that the spinel sintered at 1350 °C exhibited relatively high wear resistance with small track width and a low friction coefficient (0.35).

The results of this study also indicate that transparent ceramics were obtained with optimum transparency for sintering at 1350 °C. For this sintering temperature, a good combination of the mechanical properties was obtained. This transparent ceramic can be exploited in a wide range of applications, optical lenses and transparent armor.

**Author Contributions:** Conceptualization, K.H., S.B. (Salim Benaissa), A.C., S.B. (Sofiane Bouheroum), A.A., M.H. and G.F.; methodology, K.H., S.B. (Salim Benaissa), S.B. (Sofiane Bouheroum), M.H. and G.F.; validation, S.B. (Salim Benaissa), M.H. and G.F.; formal analysis, S.B. (Salim Benaissa), M.H. and G.F.; writing—original draft preparation, K.H., S.B. (Salim Benaissa), A.C., S.B. (Sofiane Bouheroum), M.H. and G.F.; writing—review and editing, K.H., S.B. (Salim Benaissa), M.H. and G.F.; visualization, K.H., S.B. (Salim Benaissa), M.H. and G.F.; supervision, K.H., S.B. (Salim Benaissa), M.H. and G.F.; All authors have read and agreed to the published version of the manuscript.

**Funding:** This research received no external funding.

**Institutional Review Board Statement:** Not applicable.

**Informed Consent Statement:** Not applicable.

**Data Availability Statement:** Data available upon request.

**Conflicts of Interest:** The authors declare no conflict of interest.

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
