# Peer review of "Mechanical Behavior of Transparent Spinel Fabricated by Spark Plasma Sintering"

_ceramics, doi:10.3390/ceramics6020072_

Round 1

Reviewer 1 Report

This paper presents interesting results on the influence of the sintering temperature on microstructure and mechanical and optical properties of SPS spinel samples.

Some mistakes should be corrected: i)The acronym of the lab INSA Lyon is "MATEIS" ii) avoid to use the French word "elaboration" and replace it by processing or fabrication. same remark for "elaborated". iii) Lign 203 reaches, Lign 153  capital letter for All, iv)Lign 483 no verb in the sentence. iv) Lign 285" larger grain size" suppress the "s" 

Remarks on the result presentation: i) some problems on significant number: For instance the hardness: 16.66 GPa. Are you sure that the second figure in decimal is significant? It would be better to restrict to 16.7 GPa. same remarks for the agglomerate size: 474.66 µm. and for relative density measured by Archimedes method: 99.65???I have some doubt with such high precision.  ii) For all mechanical characteristics, precise the number of assays: n=?

Materials and methods: The SPS cycle was chosen from previous studies. It would be nice to add a sentence explaining the necessity of heating rate changes: influence on density and microstructure development?

Results: Lign215 If you admit that 2 decimal figures are exaggerated for density values, there is no difference between samples sintered at 1300°C and 1350°C both 99.7%. 

Ligns 216 to 219: your explanation is not convincing. Grain growth usually does not improve the density. 

Some interrogations on the presented microstructures: The appearance of the microstructures for 1350 and 1400°C obtained by SEM analysis is strange and suggests the presence of grains of different hardness values which is not the case. Polishing defect? The polishing conditions should be adapted to each sample. For the 1300°C sample image, the magnification is different. Could you increase it to check if the surface appearance is the same as for the 2 other samples.

Contamination with carbon during the SPS: This means that no annealing thermal treatment was applied to the sintered samples? It could be interested to perform such a treatment and to compare after this, the optical transmission for the 3 samples.

Lign 282 and 283 I disagree taking into account that the density values are identical. Lign 292: "value is slighly higher that  found by Krell and Benaissa" Was the measurement method identical to yours? 

Lign 310 How can you say that the 1350°C sample presents a more homogeneous microstructure compared to 1300°C one?

Table 4: add the standard deviation to the values of E and the number of assays. (n=?). Lign 343 I do not understand the "worst friction behaviour" in the case of 1350°C sample. it is best friction , no?

Lign 356 The 1350°C sample presents the narrowest wear track width but with a very high standard deviation. Comments?

Conclusion: should be completed. No comments on the influence of the sintering temperature on the microstructure and mechanical properties. Only the data concerning 1350°C are summarized.

Author Response

Friday 26 May 2023

Subject: Detailed Response to Reviewers

Ref. No.: ID: ceramics-2407894

Title: Mechanical behavior of transparent spinel fabricated by Spark Plasma

Sintering

Editors of Journal of Ceramics,

First of all, we are very keen to thank the referees for the interest carried to our work and to all the raised remarks. You'll find in what follows the details concerning the given answers. 

Reviewer's comments and our answers:

Reviewer 1:

C1: Some mistakes should be corrected: i)The acronym of the lab INSA Lyon is "MATEIS" ii) avoid to use the French word "elaboration" and replace it by processing or fabrication. same remark for "elaborated". iii) Lign 203 reaches, Lign 153  capital letter for All, iv)Lign 483 no verb in the sentence. iv) Lign 285" larger grain size" suppress the "s".

A1: As suggested by the Reviewer, all mistakes are corrected in the revised manuscript.

C2: Remarks on the result presentation: i) some problems on significant number: For instance the hardness: 16.66 GPa. Are you sure that the second figure in decimal is significant? It would be better to restrict to 16.7 GPa. same remarks for the agglomerate size: 474.66 µm. and for relative density measured by Archimedes method: 99.65???I have some doubt with such high precision.  ii) For all mechanical characteristics, precise the number of assays: n=?

A2: The recommendation of the Reviewer is taken into account and all values presented in the first figure in decimal, except for friction coefficient and relative density witch the second figure in decimal is significant, because all the values of the latter is greater than 99 %. Moreover, we have introduced the number assays of mechanical characteristics.

C3: Materials and methods: The SPS cycle was chosen from previous studies. It would be nice to add a sentence explaining the necessity of heating rate changes: influence on density and microstructure development?

A3: As suggested by the Reviewer, we have introduced a sentence witch explaining the necessity of heating rate changes: influence on density and microstructure development.

C4: Results: Line215 If you admit that 2 decimal figures are exaggerated for density values, there is no difference between samples sintered at 1300 °C and 1350 °C both 99.7%. 

A4: we mentioned the values of the relative density with the second figure in decimal because all the values of the latter are greater than 99% ,and we mentioned that the density increased slightly with increasing sintering temperature in the revised version .

C5: Lines 216 to 219: your explanation is not convincing. Grain growth usually does not improve the density.

A5: The recommendation of the Reviewer is taken into account and lines 216 to 219 has been improved.

C6: Some interrogations on the presented microstructures: The appearance of the microstructures for 1350 and 1400°C obtained by SEM analysis is strange and suggests the presence of grains of different hardness values which is not the case. Polishing defect? The polishing conditions should be adapted to each sample. For the 1300°C sample image, the magnification is different. Could you increase it to check if the surface appearance is the same as for the 2 other samples.

A6: The polishing of our samples was carried out under the same conditions, thorough polishing was required using an automatic polisher (Mecatech 334). Several steps were necessary. In the first (for 60 seconds), we used diamond with a grain size of 40 µm fixed on a support. A force of 3 daN was applied. The speed of rotation of the plate Vp is equal to 300 rpm and a speed of the head of the machine of Vt 150 rpm. Then we used an abrasive support (Ï•20 µm) with the same parameters (F, Vp, Vt) for 120 s. In the third step, we used a 6 µm and 3 µm diamond suspension with a flow rate of 0.3 ml/30s, F (4 daN), Vp (300 rpm) and Vt (135 rpm) for 300 sec. Finally, we used a 1 μm abrasive slurry with the same parameters for 120s and we finished the operation with a diamond abrasive slurry of about 0.25 μm at 60s. After each polishing step, an ultrasonic cleaning operation is necessary. At the end of the polishing procedure, the thickness of the discs is reduced to approximately 1 mm. The results of the AFM confirm that our samples present a roughness of nano metric order and the specular reflection due to the roughness of surface is almost the same one for all the samples.

For the defects that appeared on the images of 1350 ° C and 1400 ° C is perhaps only caused by the chemical treatment that we carried out before the observation at the SEM as shown by Benaissa et al. [26].

C7: Contamination with carbon during the SPS: This means that no annealing thermal treatment was applied to the sintered samples? It could be interested to perform such a treatment and to compare after this, the optical transmission for the 3 samples.

A7: As suggested by the Reviewer, we have tried to improve the line transmission by annealing thermal treatment, but the change is not significant as shown by Zegadi et al.[3]. Example:

Figure : The Real In-line Transmission spectra of samples sintered by SPS at 1350°C before and after annealing thermal treatment.

C8:  -Line 282 and 283 I disagree taking into account that the density values are identical.

      -Lign 292: "value is slighly higher that found by Krell and Benaissa" Was the measurement method identical to yours?.

A8: -We agree well with the Reviewer that the density values are not identical but all values are greater than 99 %, and the difference in hardness values attributed mainly to the variation in the grain size.

Yes the measurement method identical to Krell and Benaissa, we found hardness values are convergent but there is a difference.

C9: Lign 310 How can you say that the 1350 °C sample presents a more homogeneous microstructure compared to 1300 °C one?

A9: The sample heated at 1350 °C exhibits a more homogeneous microstructure compared to the one heated at 1300°C. This is attributed to the grain growth phenomenon, which plays a crucial   role   in   promoting   a   greater   homogeneity.   As   the   grains   continue   to   grow,   they gradually occupy the empty spaces between neighboring grains, resulting in a uniform and less heterogeneous microstructure.

C10:        - Table 4: add the standard deviation to the values of E and the number of assays. (n=?).

Lign 343 I do not understand the "worst friction behaviour" in the case of 1350 °C sample. It is best friction , no?

A10: -For the elastic modulus we made a single measurement, since the ultrasound spectrum is the same for the same sample.

We agree well with the Reviewer that the sample sintered at 1350 °C exhibits the best tribological behavior and we are corrected in the revise paper.

C11: Lign 356 The 1350 °C sample presents the narrowest wear track width but with a very high standard deviation. Comments?

A11:  We agree well with the Reviewer that the sample sintered at 1350°C exhibits a very high standard deviation. This may be due to surface flatness defects or the ball is not cleaned properly during the test.

C12: Conclusion: should be completed. No comments on the influence of the sintering       temperature on the microstructure and mechanical properties. Only the data concerning 1350°C are summarized.

A12: The recommendation of the Reviewer is taken into account and the influence of the sintering       temperature on the microstructure and mechanical properties has been introduced in the revised version.

Reviewer 2 Report

This manuscript reported on the mechanical properties of a transparent nanostructured ceramic magnesium aluminate spinel (MgAl2O4) was fabricated by Spark Plasma Sintering (SPS). The research content is interesting and the manuscript is relatively well organized. The size of the font used for the graphs is small, so it would be nice to make it bigger so that it can be distinguished well. It is required to present a more detailed model as to why the hardness increases.

Author Response

Reviewer 2:

C1: The size of the font used for the graphs is small, so it would be nice to make it bigger so that it can be distinguished well.

A1: The recommendation of the Reviewer is taken into account and the authors has been increased the font size in the revised manuscript.

C2: It is required to present a more detailed model as to why the hardness increases.

A2: As suggested by the Reviewer, the authors have improved and presented a more detailed model as to why the hardness increases in the revised manuscript.

Reviewer 3 Report

The authors have studied mechanical behavior of transparent spinel (MgAl2O4) fabricated by spark plasma sintering. The research is well designed but presented not very clearly. A good comparative analysis of existing publications concerning the tasks set in the work is performed. The methodological section of the manuscript is presented in sufficient detail but some issues should be explained. The authors used the modern equipment for tests of samples as well as visualization and assistance in the interpretation of the obtained results. The microstructural, optical, and mechanical properties of the sintered samples were characterized using SEM, AFM, spectrophotometer with an integrating sphere, instrumented Vickers indenter, Pin-on-Disk tribometer, scratch tester, and sandblasting device. The authors found that the sintering temperature has a great influence on the characteristics of the elaborated spinel samples.

However, some shortcomings should be corrected to make the manuscript acceptable for publication in Ceramics.

(1) Instead of a single word “toughness”, the authors should use the phrase “fracture toughness” throughout the manuscript.

(2) Line 110: In the caption to Table 1, the phrase “This is a table” should be removed. In addition, the table seems to represent the remaining impurities, not chemical composition of MgAl2O4 spinel powder (see, for example, https://doi.org/10.1179/1743285511Y.0000000007 ). The effects (or absence of the effects) of these impurities should be discussed in the manuscript.

(3) Lines 141–142: The authors should add a reference from where Formula (2) for Vickers hardness calculation was taken. Besides, decimals should be checked in accordance with units.

(4) Lines 146–148: Formula (3) is given with a mistake. The authors should check it according to the equation of Anstis et al. [29]. In addition, they must justify whether this is a simple typographical error or whether all the values obtained are the result of this incorrect formula.

(5) The indentation load of 5N for measurement of fracture toughness (KIC) of the spinel by the indentation technique using the equation of Anstis et al. [29] should be substantiated. As a reference, the authors can use the explanation given in https://doi.org/10.3390/ma15082707 for zirconia based ceramics.

(6) In Figure 15, the scratching direction should be indicated.

In my opinion, English language of this manuscript should be improved.

Author Response

Reviewer 3:

C1: Instead of a single word “toughness”, the authors should use the phrase “fracture toughness” throughout the manuscript.

A1:  The recommendation of the Reviewer is taken into account and the authors has been replaced a single word “toughness” by the phrase “fracture toughness” in the revised manuscript.

C2: Line 110: In the caption to Table 1, the phrase “This is a table” should be removed. In addition, the table seems to represent the remaining impurities, not chemical composition of MgAl2O4 spinel powder (see, for example,  https://doi.org/10.1179/1743285511Y.0000000007 ).  The effects (or absence of the effects) of these impurities should be discussed in the manuscript.

A2:  The recommendation of the Reviewer is taken into account and the authors have  replaced the phrase “chemical composition of MgAl2O4 spinel powder” by “Impurities in chemical composition of the used MgAl2O4 spinel powder.”, and It is worth noting that the presence of these minor impurity elements does not have any discernible impact on the properties of transparent spinel.

C3: Lines 141–142: The authors should add a reference from where Formula (2) for Vickers hardness calculation was taken. Besides, decimals should be checked in accordance with units.

A3: As suggested by the reviewer, we have added the reference from where Formula (2) for Vickers hardness and we have checked the decimals in accordance with units.

C4: Lines 146–148: Formula (3) is given with a mistake. The authors should check it according to the equation of Anstis et al. [29]. In addition, they must justify whether this is a simple typographical error or whether all the values obtained are the result of this incorrect formula.

A4: The recommendation of the reviewer is taken into account and the authors have  corrected the two mistakes in the revised manuscript and we have checked that the correct formula is used in our work.

C5: The indentation load of 5N for measurement of fracture toughness (KIC) of the spinel by the indentation technique using the equation of Anstis et al. [29] should be substantiated. As a reference, the authors can use the explanation given in https://doi.org/10.3390/ma15082707 for zirconia based ceramics.

A5: As suggested by the reviewer, we have introduced that the equation of Anstis et al. [31], which recently concluded that this formula presented the best fits of the characterization of the ceramics [32]:

C6: In Figure 15, the scratching direction should be indicated.

A6: The recommendation of the reviewer is taken into account and the authors have been indicated the scratching direction for Figure 15 in the revised manuscript.

C7: English language of this manuscript should be improved.

A7: As suggested by the reviewer, we have improved the english language of this manuscript.

Round 2

Reviewer 3 Report

All the reviewer’s comments were taken into account by the authors. The manuscript can now be accepted for publication in Ceramics.

English language of this manuscript should be slightly improved.